# The Health-Related Fatty Acid Profile of Milk from Holstein–Friesian Cows as Influenced by Production System and Lactation Stage

**DOI:** 10.3390/ani14233492

**Published:** 2024-12-03

**Authors:** Zenon Nogalski, Martyna Momot, Monika Sobczuk-Szul, Anna Nogalska

**Affiliations:** 1Department of Animal Nutrition, Feed Science, and Cattle Breding, Faculty of Animal Bioengineering, University of Warmia and Mazury in Olsztyn, Oczapowskiego 5, 10-719 Olsztyn, Poland; martyna.momot@uwm.edu.pl (M.M.); monika.sobczuk@uwm.edu.pl (M.S.-S.); 2Department of Agricultural and Environmental Chemistry, Faculty of Agriculture and Forestry, University of Warmia and Mazury, ul. Oczapowskiego 5, 10-719 Olsztyn, Poland; anna.nogalska@uwm.edu.pl

**Keywords:** bovine milk, CLA, conventional farms, fatty acids, milk, organic farms, proximate composition

## Abstract

Organic milk is known to be healthier for consumers, but is this true throughout lactation? Does the negative energy balance (NEB), observed in intensively fed cows in conventional herds (CDHs) during the first stage of lactation, significantly alter the fatty acid profile of milk? In the present study, milk yield decreased (the decrease was more pronounced in organic herds, ODHs) and the concentrations of protein and dry matter increased with advancing lactation. An analysis of fat from milk samples collected in winter (preserved feed) confirmed that organic milk is healthier for consumers, but it also has a higher somatic cell count than conventional milk. The fatty acid profile of milk fat was more desirable in ODHs than in CDHs (higher concentrations of polyunsaturated fatty acids (PUFAs), including n-3 PUFAs, trans-vaccenic acid, linolenic acid, and conjugated linoleic acid, and a higher desaturase index).

## 1. Introduction

Milk has been a part of the human diet for 8000 to 10,000 years, ever since aurochs, the ancestors of modern cattle, have been domesticated in Eurasia. The knowledge about milk’s health-promoting properties increased over time, and milk became a universally recognized source of nutrients [1]. The consumption of milk and dairy products has a beneficial influence on bones, the circulatory system, and the gastrointestinal microbiome [2,3,4]. Milk consumption is associated with reduced stunting, and the World Health Organization (WHO) recommends that 25–33% of dietary protein should come from dairy products in malnourished children [3]. Milk components decrease the risk of osteoporosis and cardiovascular diseases [4]. Modern consumers tend to avoid highly processed and intensively produced foods due to the lower nutritional quality of such products as well as the environmental impact of conventional farming [5,6]. In organic farming, synthetic fertilizers are banned, animal operations are not allowed without land, and there are strict limits on stocking rates [7].

As a result, consumers have a growing demand for organic products which are regarded as healthier [8,9]. The above applies particularly to organic milk, which is more abundant in health-promoting compounds and has a more desirable fatty acid (FA) profile than milk from conventional farms [10]. Organic milk contains more polyunsaturated fatty acids (PUFAs), including omega-6 and omega-3 PUFAs, than conventional milk [8]. In different farming systems (organic vs. conventional), nutrition is associated with animal welfare [11]. In organic farms, cows are fed forage that is grown without pesticides, and farms have to comply with European Union regulations on organic production standards and the use of pharmaceuticals [12]. Limited antibiotic use can increase the prevalence of mastitis in organic livestock herds [13]. However, this is not always the case. Some studies have shown that mastitis was less prevalent in adequately managed organic herds than in conventional herds [14,15]. Organic farms are generally characterized by lower milk yields, but the produced milk is more nutritious and has a more desirable FA profile [16]. Milk fat is composed of more than 400 FAs that are partly synthesized in the mammary gland (nearly 50%), provided by feed (that undergoes biohydrogenation in the rumen), and mobilized from body fat reserves in a period of negative energy balance (NEB) [17,18]. Fatty acids are generally classified into two groups: short-chain FAs (SCFAs) and medium-chain FAs (MCFAs) (C4–C15) that are synthesized de novo in the mammary gland, and long-chain FAs (LCFAs, C17-C26) that are derived from feed [19]. Medium-chain FAs (C16) are both synthesized in the mammary gland and supplied in the diet [20]. Some milk FAs, including conjugated linoleic acid (CLA) and n-3 polyunsaturated FAs (PUFAs), deliver health benefits by decreasing the risk of cancer and diabetes [21,22]. The quality and quantity of saturated FAs (SFAs) and some unsaturated FAs (UFAs) can be evaluated using the atherogenic index (AI) [23]. Milk fat can also be classified based on the desaturase index (DI) and the n-6/n-3 ratio. The n-6/n-3 ratio denotes the proportions of n-6 and n-3 FAs that exert antagonistic effects on physiological processes and are essential for healthy growth and development. Previous research has shown that lowering the dietary n-6/n-3 ratio can reduce the risk of many chronic diseases [24].

Numerous studies have been conducted to determine whether the FA composition of cow’s milk can be modified to provide greater health benefits to consumers [25]. The influence of various exogenous and endogenous factors on the FA profile of milk was also examined [26,27]. Most researchers investigated the effect of nutritional factors and focused mainly on the first few weeks after calving. However, the influence of lactation stage (LS) on the composition of milk fat has been rarely studied, and the available reports are often inconclusive [20,28,29]. Feed intake changes in different stages of lactation, which affects the content of enzymatic and microbial flora, rumen capacity, rumen passage rates, rumen isomerization, biohydrogenation, the activity of stearoyl-CoA desaturase (SCD) in the mammary gland, and energy balance in dairy cows [30,31]. At the beginning of lactation, body fat reserves in high-yielding cows are mobilized due to NEB, which leads to the release of LCFAs (mainly stearic and oleic acids) from adipocytes [32]. According to Stoop et al. [18], LS and the corresponding energy balance contribute significantly to the variation in milk fat composition by modifying the activity of FA pathways. The research hypothesis postulates that cows kept in organic herds and fed extensively with organic feed will produce less milk while remaining more stable during lactation than intensively fed cows that are at higher risk of NEB in the first stage of lactation. Therefore, this study aimed to evaluate the effect of production system and LS on the yield, centesimal composition and fatty acid profile of milk from Holstein–Friesian cows.

## 2. Materials and Methods

### 2.1. Animals

The study was conducted on 14 herds of dairy Holstein–Friesian cows, including six certified organic dairy herds (ODHs) and eight conventional dairy herds (CDHs). All farms are located in Warmian–Masurian Voivodeship (northeastern Poland). The average number of cows in CDHs was 52, and in ODHs, it was 44. The average annual milk yield in CDHs was 9600 kg, and in ODHs, it was 6200 kg. In both production systems, cows were kept in tie-stall and free-stall barns, on shallow litter. In tie-stall barns with straw bedding, manure was removed mechanically twice daily before milking (with a chain scraper or manually), and the cows were milked twice daily (pipeline milking machine). In free-stall barns with straw bedding, manure was removed mechanically once daily in the morning before feeding (using a tractor-mounted scraper), and the cows were milked twice daily (auto-tandem fishbone milking parlor).

The animals were fed maize silage, haylage, meadow hay, and concentrates. The proximate chemical composition and FA profile of diets are presented in Table 1 and Table 2.

The ingredients (% dry matter) and chemical composition of diets (means for all herds) are shown in Table 3. The two production systems (CDHs and ODHs) differed in the types and proportions of feedstuffs in the ration. The diets were based on maize silage and a high proportion of concentrate in CDHs, and on haylage, meadow hay, and a low proportion of concentrate in ODHs. The concentrate included cereal grains, maize, legume meal, and rapeseed meal.

The procedures laid down in Council Regulation [12] on organic production and the labeling of organic products and the detailed rules laid down in Commission Regulation [7] were strictly observed in certified organic farms. A total of 539 milk samples were collected twice over the winter season in January (during morning milking) and in March (during evening milking). The samples were collected randomly from up to 30% of cows at a given stage of lactation (7–45, 46–90, 91–135, 136–180, 181–225, 226–270, 271–315, and 316–360 days after calving). Cows were selected in particular stages of lactation according to the order of calving. Animals in all herds included in the study were under veterinary care. None of the cows showed clear clinical symptoms of mastitis or was treated for mastitis. Approximately 100 mL milk samples were collected during milking using separators. Milk samples were collected in glass containers with 1 mL (20% wt/vol) of 2-bromo-2-nitropro-pane-1,3-diol (bronopol; VWR International AB, Stockholm, Sweden). Milk samples were transported at refrigeration temperature to the laboratory of the Department of Animal Nutrition, Feed Science, and Cattle Breding at the University of Warmia and Mazury in Olsztyn. Two portions of milk were taken from the bulk sample; the first portion was stored at 4 °C and was used to determine milk composition and the somatic cell count (SCC), and the other portion was stored at −20 °C and was used to analyze the FA profile of milk fat. Daily milk yield was determined for each cow, and it was converted to energy-corrected milk (ECM): milk with standardized energy content [33].
(1)$$ECM\left(kg\right)=milk(kg)\cdot \frac{\left(0.383\cdot fat\left(\%\right)+0.242\cdot protein\left(\%\right)+0.7832\right)}{3.140}$$

The body energy reserves of dairy cows were estimated by an experienced evaluator based on body condition scores (BCSs) using a 5-point scale with 0.25-point increments where 1 denotes a very thin cow and 5 denotes an excessively fat cow [34].

### 2.2. Analysis of Milk Composition and Fatty Acid Profile

Fresh milk samples were analyzed to determine their chemical composition (crude protein, casein, fat, lactose, urea, and dry matter content) by infrared spectrophotometry using the MilkoScan FT 120 (FossElectric, Duisburg, Germany) and SCC by flow cytometry using the BactoCount IBC (Bentley, Chaska, MN, USA). In addition, the fat/protein ratio was calculated.

Milk fat was extracted by the Röse–Gottlieb method according to PN-EN ISO [35]. The proportions of 41 FAs in milk fat were determined by gas chromatography, using the Varian CP 3800 system (EquipNet, Inc., Canton, MA, USA) with a split/splitless injector and a flame-ionization detector (FID). Samples (1 μL) of FA methyl esters were placed on a CP-Sil 88 capillary column (length: 100 m, inner diameter: 0.25 mm). The results were processed using the GALAXIE Chromatography Data System. Fatty acids were identified by comparing their retention times with those of commercially available reference standards purchased from Supelco, Inc. (Sigma Aldrich, Bellefonte, PA, USA). Analyses of samples and reference standards were performed under identical conditions, i.e., carrier gas—helium, injector temperature −260 °C, detector temperature −260 °C, initial oven temperature −110 °C, maintained for 5 min, then increased to 240 °C at a rate of 3 °C/min, maintained at 240 °C for 10 min, then increased to 249 °C at a rate of 1 °C/min. The total time of a single analysis was 68 min. Saturated FAs (SFAs), unsaturated FAs (UFAs)—including monounsaturated FAs (MUFAs) and polyunsaturated FAs (PUFAs)—were expressed as the relative percentages of total FAs, and the n-6/n-3 PUFA ratio was calculated. The atherogenicity index (AI) = [12:0 + 4 (14:0)+16:0]/[MUFA+ PUFA] and the thrombogenicity index (TI) = (14:0 + 16:0 + 18:0)/[(MUFA + n6 PUFA)/2 + 3 (n3 PUFA) + (n3 PUFA/n6 PUFA)] were also determined [22]. The health-promoting index (HPI) was calculated according to Chen et al. [36] based on the concentrations of MUFAs, PUFAs, and other FAs: HPI = (n-6PUFA+n-3PUFA +MUFA)/[(C12:0 + (4 × C14:0) + C16: 0)]. The values of the HPI ranged from 0.16 to 0.68. Dairy products with a high HPI are assumed to be more beneficial to human health. The DI was determined using the formula proposed by Garnsworthy et al. [37]: DI = (C14:1 cis-9 × 100)/(C14:0 + C14:1 cis-9).

### 2.3. Data Analysis

The results were processed in the Statistica 13.3 program [38]. In order to obtain the normal distribution of a variable, the SCC was log-transformed according to the formula:(2)Y=Ln(x)
where

*x*—SCC determined in milk samples.

The remaining variables had a normal distribution. The least squares method was used to determine whether milk composition and FA profile were affected by production system (conventional vs. organic) and LS (7–45, 46–90, 91–135, 136–180, 181–225, 226–270, 271–315, and 316–360 days of lactation). To conduct the analysis, the following model was created:Y*_ijk_* = μ + A*_i_* + B*_j_* + (AB)*_ij_* + e*_ijk_*(3)
where Y*_ijk_* is the analyzed parameter, μ is population mean, A*_i_* is the effect of the production system (1, 2), B*_j_* is the effect of LS (1–8), (AB)*_ij_* is the production system x LS interaction, and e*_ijk_* is random error.

The significance of differences between means was determined by Tukey’s test. The significance level was set at *p* < 0.05 and *p* < 0.01.

## 3. Results

The average daily milk yield was 11.4 kg higher (*p* < 0.01) in CDHs than in ODHs (Table 4). The concentrations of milk constituents, including fat, protein, lactose and, consequently, dry matter, were also higher in CDHs than in ODHs. Lactation stage had a significant effect on milk yield as well as the protein and dry matter content of milk. Daily milk yield peaked on lactation days 46–90. In the later stages of lactation, daily milk yield decreased, and the concentrations of protein and dry matter in milk increased. The average difference between the peak daily milk yield and milk yield on lactation days 316–360 reached 16.4 kg of milk, which means a decrease of over 53%.

An interaction between production system and LS was found for daily milk yield (Figure 1). The decrease in daily milk yield in the first stage of lactation was greater in ODHs than in CDHs, and lactation peak was not observed in ODHs. The SCC was higher (by 96,400 cells on average) in organic milk than in conventional milk (*p* < 0.01). Milk urea levels were higher (*p* < 0.05) in CDHs than in ODHs. The BCSs of cows (assessing body fat and muscle reserves) were also higher in CDHs. An interaction between production system and LS was noted for BCSs (Figure 2). In CDHs, cows mobilized their body fat reserves in the first stage of lactation, which was reflected in their BCSs (Figure 2).

The SCC increased with advancing lactation, but the differences were not significant. Significant changes in BCSs were observed during lactation, and the lowest value of this parameter was recorded 46–90 days after calving. The production system affected the proportions of SCFAs and PUFAs (including n-3 PUFAs) and the n-6/n-3 PUFA ratio in milk (Table 5). The proportion of SCFAs (*p* < 0.05) and the n-6/n-3 PUFA ratio (*p* < 0.01) were higher in milk produced in CDHs. Milk fat contained more PUFAs (*p* < 0.01), including n-3 PUFAs (*p* < 0.01), in ODHs than in CDHs. The difference in n-3 PUFA-s in favor of ODHs was 0.18, which was 31%.

The proportion of SCFAs was affected by LS, and it decreased with advancing lactation. The values of DI, including desaturase activity, the CLA/vaccenic acid ratio (*p* < 0.01) and the oleic acid/stearic acid ratio (*p* < 0.05), were higher in organic milk.

An interaction between production system and LS was found for the proportions of SCFAs (Figure 3) and MUFAs (Figure 4) in milk fat. In CDHs, the proportion of MUFAs in milk fat varied widely during lactation, whereas in ODHs, it remained stable and increased only in the last stage of lactation. An interaction between the experimental factors was also noted for the HPI (Figure 5).

The production system influenced the proportions of some major FAs in total FAs (Table 6). The content of trans-vaccenic acid (TVA), linolenic acid (LNA), CLA, and eicosapentaenoic acid (EPA) in milk fat was significantly higher in ODHs than in CDHs. In CDHs, cows produced milk with higher (*p* < 0.05) concentrations of oleic acid (OA) and butyric acid (BA). The BA content of milk fat was also affected by LS, and it decreased as lactation progressed. The average difference between the beginning and end of lactation reached 0.19 g, which means a decrease of over 7%. An analysis of three SFAs (lauric, myristic and palmitic acids) revealed that the proportion of palmitic acid (C16:0) in total FAs in milk fat was higher (*p* < 0.05) in ODHs (Table 7). The proportion of palmitic acid in milk fat remained relatively stable (around 30%) throughout lactation in ODHs, and it decreased with advancing lactation in CDHs.

## 4. Discussion

The intensive feeding of cows in CDHs resulted in higher milk production compared with CDHs, which corroborates the findings of other authors [15,39]. In the current study, the difference in daily milk yield between the analyzed systems reached 11.4 kg, and Barłowska et al. [40] reported that Holstein–Friesian cows in the conventional system produced on average 6.1 kg more milk than their counterparts in the organic system. The higher milk yield in conventional farms is the result of both more intensive feeding and optimally balanced diets compared with organic farms. Król et al. [41] demonstrated that the nutrient requirements of dairy cows were not fully met in the organic production system. Inadequate coverage of the nutritional needs of cows in organic herds is also the main reason for lower concentrations of milk constituents relative to conventional herds [42]. According to Zagorska and Ciprovica [43], the lower protein content of organic milk is due to the lower starch content of feed. In conventional herds, cows are mainly fed maize silage, which stimulates bacterial protein synthesis in the rumen and has a positive effect on milk protein content [44]. This explains why the protein content of milk was higher in CDHs fed maize silage in the present study. Fat is the main source of energy in milk, and it is the most easily digestible animal fat in the human diet. Changes in the content and composition of milk fat are largely determined by the content, composition, and form of crude fiber as well as the starch and sucrose content of the cow’s diet [7].

The results of studies investigating the differences in fat content between organic and conventional milk are inconclusive. Zagorska and Ciprovica [43] found that organic milk had higher fat content than conventional milk, whereas Sundberg et al. [45] and Barłowska et al. [40] observed higher fat percentage in conventional milk. In the present study, milk fat content was lower in ODHs, which could be due to less abundant feeding and the absence of dietary supplements. In conventional dairy farming, high-yielding cows are commonly fed in excess of requirements using diets supplemented with high amounts of energy and protein concentrates. However, in organic dairy farming, the aim is to optimize available resources rather than maximize production, so that in most cases, systems are based on the maximum use of forage [46]. Lactose is the main carbohydrate in milk that maintains its osmolarity and is positively correlated with milk yield [10]. Zagorska and Ciprovica [43] noted differences in milk lactose content between organic and conventional production systems and attributed them to different diets. In the current experiment, changes in milk lactose concentrations also resulted from differences in feeding regimes between ODHs and CDHs.

The SCC, which is strongly associated with udder health, tends to be higher in organic farms where the use of antibiotics to treat mastitis is highly limited [12,45]. In the present study, the compliance with organic production standards made it difficult to maintain adequate udder health. However, in some studies, the proportion of cows with a high SCC was lower in organic herds than in conventional herds, most likely due to lower milk production in the former, which translates into lower stress on the cow’s body and lower susceptibility to inflammation [13,14]. In this study, lower milk urea levels and lower body fat and muscle reserves (BCS) in ODHs pointed to the lower protein content of the feed ration. Intensive feeding and the high genetic potential of dairy cows are determinants of high milk yield [47]. The first few months after calving are considered critical because in this period, feed intake (especially dietary energy intake) is insufficient to adequately meet the increasing needs of lactating cows. In the current experiment, cows in CDHs, characterized by high milk production in the second and third months of lactation, mobilized their body fat reserves in response to the energy deficit (Figure 2), which affected the course of lactation (Figure 1). The physiological decline in milk yield as lactation progressed was greater in cows in ODHs than in CDHs, because the former were fed less intensively and did not mobilize body fat reserves to support lactation. In CDHs, intensive feeding and decreasing BCS stimulated milk production for a long time after calving. In ODHs, lactation peak did not occur due to less intensive feeding, and daily milk yield decreased relatively quickly. In the later stages of lactation, the lower productive performance of cows was associated with increasing concentrations of milk constituents, in particular protein and fat, while the opposite was observed for lactose content. Milk fat concentration decreases after calving, reaching the lowest level 40 to 60 days post-partum. This decline is mostly due to fat dilution as milk yield increases and the production of lactose by the mammary gland increases as well [48]. In the present study, the SCC increased with advancing lactation, which could contribute to the decrease in milk production. According to Gonçalves et al. [49], an increase in the SCC is accompanied by a decrease in milk yield, especially when the SCC exceeds the threshold of 200,000 cells per mL of milk.

The fact that dairy cows mobilize their body fat reserves in early lactation to compensate for the energy deficit was confirmed by changes in BCS during lactation [47]. In this experiment, the difference between the highest and lowest average BCS of cows in CDHs reached 0.7 points, indicating that some of the milk FAs could come from the mobilized body fat reserves. However, the main source of milk FAs are lipids supplied with feed, which undergo hydrolysis followed by biohydrogenation from UFAs (mainly C18:3, C18:2) to SFAs (mainly C18:0) [50]. The type of FAs that reach the rumen varies depending on the diet. Roughage is a rich source of linoleic acid (C18:2) and linolenic acid (C18:3) [51], while cereals provide oleic acid (C18:1) and linoleic acid (C18:2) [52]. The diets fed to cows in ODHs were based on haylage, which is rich in linolenic acid (44.51%), and the diets fed to cows in CDHs were based on ground grain and maize silage, which are rich in oleic acid (25.32% and 36.35%, respectively) and linoleic acid (46.38% and 27.63%, respectively). A comparison of the FA profiles of organic milk and conventional milk revealed differences in favor of organic milk, especially with respect to nutritionally important FAs. These differences were due to differences in the diets, mainly in fiber content, which was higher in ODHs (28.47%) than in CDHs (19.17%). When comparing the FA profile of milk in both production systems and its impact on consumer health, milk fat content (4.025 in CDHs and 3.86% in ODH) should be considered. For example, taking into account the differences in milk fat content, CLA concentration (0.62% in CDHs and 0.95% in ODHs) is 0.249 g and 0.366 g per L milk in CDHs and ODHs, respectively, which favors organic milk.

In the present study, the DI was higher in organic milk. This index estimates the activity of stearoyl-coenzyme-A desaturase (SCD)/Δ9-desaturase by comparing product-to-precursor FA ratios, and it is important for the production of CLA and MUFAs [37]. Similarly to the work of Benbrook et al. [53] and Średnicka-Tober et al. [54], organic milk analyzed in this study contained more PUFAs, including n-3 PUFAs, than conventional milk. Ellis [55] demonstrated that the n-6/n-3 PUFA ratio was more than 60% lower in organic milk than in conventional milk. In the current experiment, the n-6/n-3 PUFA ratio was 21% lower in organic milk than in conventional milk, and it was closer to the optimal n-6:n-3 FA ratio in the human diet of 1:1 [1]. However, it should be stressed that Ellis [55] analyzed milk collected throughout the year, whereas in the present study, milk samples were collected in winter. The beneficial influence of fresh pasture forage on the FA profile of milk has been widely documented [9,10,56]. In this study, cows in all farms were fed preserved feed, and the type and composition of diets influenced the FA profile of milk. Haylage, which dominated the diet of cows in ODHs, had a more beneficial effect on health-related FAs in milk than maize silage [57]. A high proportion of non-structural carbohydrates (mainly starch) contributes to a decrease in the counts and activity of cellulolytic bacteria, leading to a decrease in ruminal pH [30]. In addition, in intensive milk production systems, roughage is often excessively ground, which decreases the adhesion of bacterial cells to feed particles, thus reducing the time of exposure to bacterial enzymes [50]. Most high-yielding, intensively fed cows are affected by NEB in early lactation due to the high energy requirements for milk synthesis that are not adequately met by the diet. When NEB occurs, SMCFAs (mainly C14:0) are mostly synthesized de novo in the mammary gland, and their proportion in milk fat decreases. In turn, LCFAs (C18:0 and C18:1 cis-9) are usually released from body fat reserves (mainly subcutaneous fat) during NEB, and their proportion in total FAs increases [58]. In the current study, the above relationship was observed for the concentration of oleic acid (C18:1 cis-9), which was highest in milk samples collected 7–45 days after calving, i.e., the period associated with increased risk of NEB in some cows, while the proportion of SCFAs was higher in early lactation in both organic and conventional production systems. The proportion of oleic acid in milk was on average 1.19% higher in CDHs than in ODHs. When both production system and LS were considered, the greatest difference (nearly 4%) was noted in milk samples collected 181–225 days after calving (Figure 5), which is a period usually characterized by a stable metabolic status.

## 5. Conclusions

Cows in CDHs produced more milk with higher concentrations of constituents compared with cows in ODHs. Milk produced in organic farms in accordance with the relevant standards had higher SCC, but it was also characterized by a more desirable FA profile (higher proportion of MUFAs, lower n-6/n-3 PUFA ratio). The proportions of different FA groups in milk fat were slightly modified by LS; only the proportion of SCFAs in total FAs decreased with advancing lactation. An interaction between production system and LS was found for daily milk yield, BCS, the proportions of SCFAs and MUFAs, and the HPI, which was due to the differences in feeding intensity between the analyzed systems. The changes in the FA profile of milk due to NEB in some cows in CDHs were smaller than expected.

## Figures and Tables

**Figure 1 animals-14-03492-f001:**
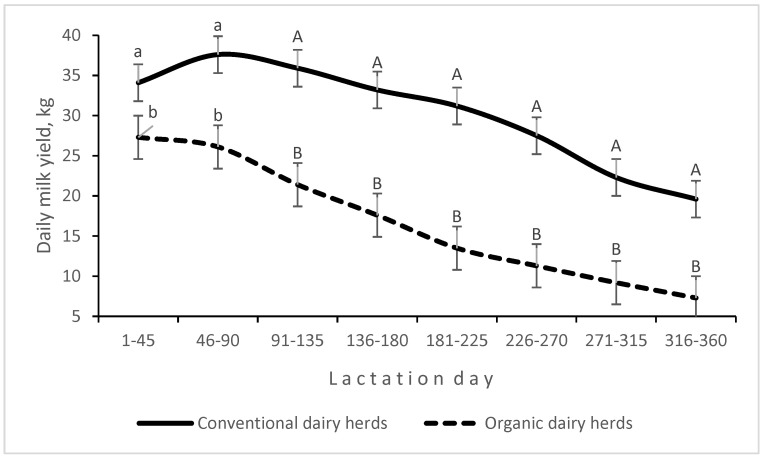
Interaction effect of production system and lactation stage on daily milk yield (kg milk); A, B (*p* ≤ 0.01); a, b (*p* ≤ 0.05).

**Figure 2 animals-14-03492-f002:**
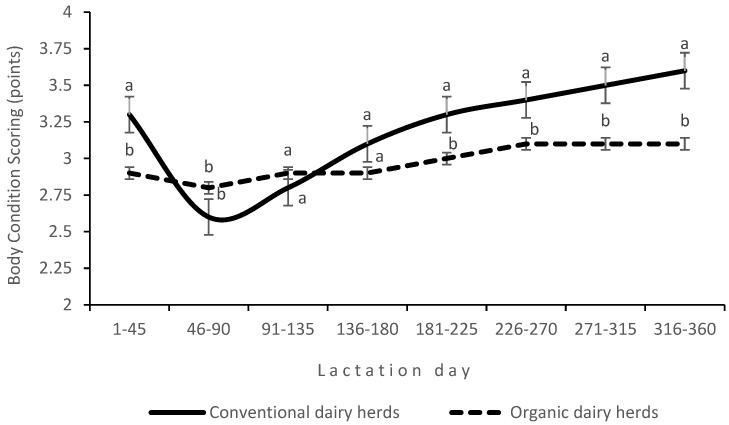
Interaction effect of production system and lactation stage on the body condition score (BCS) of cows; a, b (*p* ≤ 0.05).

**Figure 3 animals-14-03492-f003:**
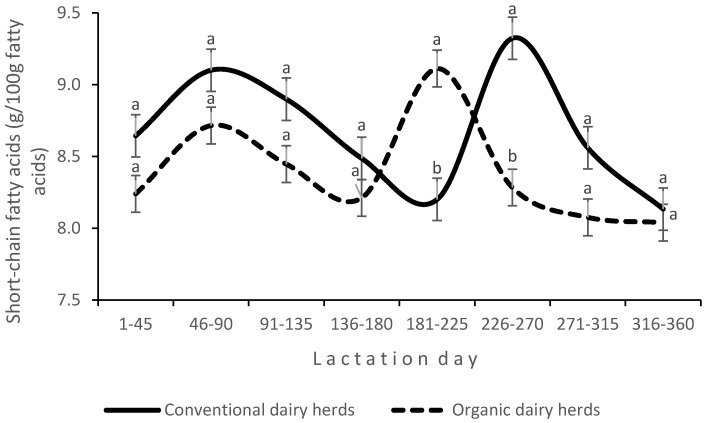
Interaction effect of production system and lactation stage on the concentrations of short-chain fatty acids in milk; a, b (*p* ≤ 0.05).

**Figure 4 animals-14-03492-f004:**
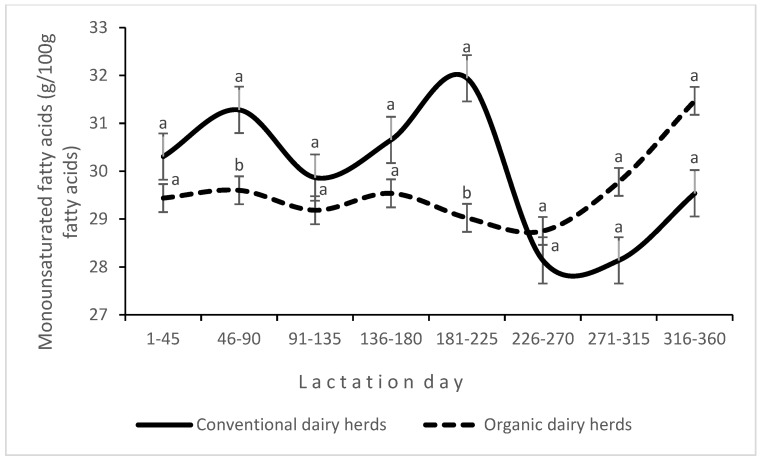
Interaction effect of production system and lactation stage on the concentrations of monounsaturated fatty acids in milk; a, b (*p* ≤ 0.05).

**Figure 5 animals-14-03492-f005:**
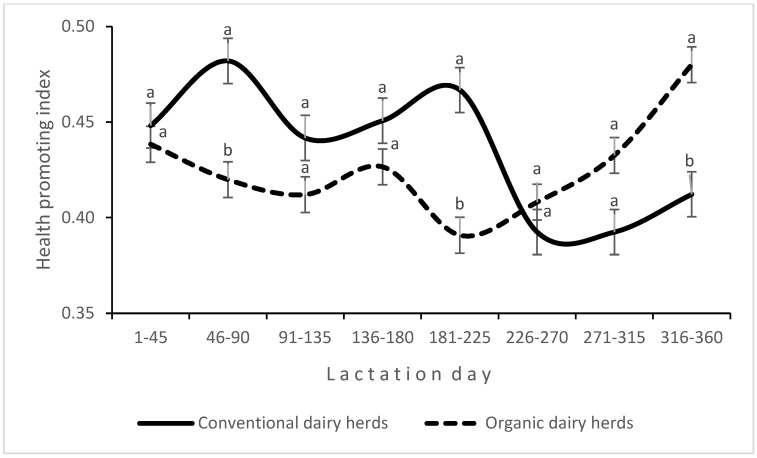
Interaction effect of production system and lactation stage on the health-promoting index; a, b (*p* ≤ 0.05).

**Table 1 animals-14-03492-t001:** Proximate composition and fatty acid profile of diets fed to dairy cows in conventional herds (mean ± standard error).

Item	Concentrate	Maize Silage	Haylage	Meadow Hay
Chemical composition (% dry matter)				
Dry matter	87.05 ± 0.44	37.41 ± 0.34	49.91 ± 0.33	83.51 ± 0.55
Ash	6.37 ± 0.14	3.66 ± 0.09	7.87 ± 0.19	6.12 ± 0.13
Crude protein	22.32 ± 0.26	9.06 ± 0.32	16.51 ± 0.21	8.25 ± 0.15
Crude fat	3.06 ± 0.08	3.72 ± 0.18	3.83 ± 0.16	1.63 ± 0.06
Crude fiber	8.77 ± 0.13	18.76 ± 0.43	29.93 ± 0.33	30.46 ± 0.38
Fatty acid profile (g 100 g^−1^ fatty acids)				
C14:0 (myristic acid)	0.34 ± 0.04	0.43 ± 0.03	0.31 ± 0.01	3.14 ± 0.04
C16:0 (palmitic acid)	28.51 ± 0.04	19.43 ± 0.06	21.32 ± 0.04	28.37 ± 0.08
C18:0 (stearic acid)	3.85 ± 0.08	3.19 ± 0.09	2.61 ± 0.08	7.12 ± 0.12
C18:1 c9 (oleic acid)	36.35 ± 0.27	25.32 ± 0.24	4.96 ± 0.17	19.59 ± 0.14
C18:2 (linoleic acid)	27.63 ± 0.24	46.38 ± 0.34	22.15 ± 0.24	13.32 ± 0.15
C18:3 (linolenic acid)	1.56 ± 0.05	5.36 ± 0.13	45.51 ± 0.24	24.67 ± 0.21

**Table 2 animals-14-03492-t002:** Proximate composition and fatty acid profile of diets fed to dairy cows in organic herds (mean ± standard error).

Item	Concentrate	Maize Silage	Haylage	Meadow Hay
Chemical composition (% dry matter)				
Dry matter	85.04 ± 0.44	33.21 ± 0.35	48.41 ± 0.34	82.41 ± 0.43
Ash	6.69 ± 0.18	3.94 ± 0.07	7.08 ± 0.19	7.53 ± 0.18
Crude protein	18.45 ± 0.25	8.25 ± 0.15	13.09 ± 0.22	7.18 ± 0.18
Crude fat	3.07 ± 0.12	3.52 ± 0.09	2.99 ± 0.09	1.72 ± 0.05
Crude fiber	9.02 ± 0.15	25.47 ± 0.35	32.11 ± 0.34	32.07 ± 0.36
Fatty acid profile (g 100 g^−1^ fatty acids)				
C14:0 (myristic acid)	0.33 ± 0.03	0.34 ± 0.04	0.23 ± 0.02	3.09 ± 0.04
C16:0 (palmitic acid)	27.23 ± 0.05	18.23 ± 0.05	22.63 ± 0.05	28.37 ± 0.08
C18:0 (stearic acid)	3.67 ± 0.08	3.11 ± 0.07	2.73 ± 0.07	8.11 ± 0.12
C18:1 c9 (oleic acid)	34.34 ± 0.35	22.42 ± 0.21	4.55 ± 0.17	17.36 ± 0.14
C18:2 (linoleic acid)	25.51 ± 0.27	43.13 ± 0.22	22.11 ± 0.24	13.43 ± 0.13
C18:3 (linolenic acid)	1.38 ± 0.07	4.35 ± 0.12	44.51 ± 0.27	22.65 ± 0.26

**Table 3 animals-14-03492-t003:** Ingredients (% dry matter) and chemical composition of diets.

Item	CDHs	ODHs
Ingredients (% dry matter) of diets		
Ground grain	30	10
Maize silage	40	20
Haylage	20	50
Meadow hay	10	20
Chemical composition of diets (% dry matter)		
Ash	5.56	6.50
Crude protein	14.45	11.48
Crude fat	3.34	2.85
Crude fiber	19.17	28.47

CDHs—conventional dairy herds; ODHs—organic dairy herds.

**Table 4 animals-14-03492-t004:** Effect of production system (PS) and lactation stage (LS) on the yield and composition of cow’s milk.

Item	Production System (PS)		Lactation Stage (LS) DIM								SE	*p*-Value		
	CDH	ODH	7–45	46–90	91–135	136–180	181–225	226–270	271–315	316–360		PS	LS	PS × LS
Number of milk samples	277	258	72	69	70	68	68	65	59	64				
Number of lactations	3.3	3.4	3.6	3.4	3.5	3.1	3.3	3.2	3.4	3.2	0.1	0.135	0.258	0.865
Lactation day	154.7	164.1	28.2	67.9	111.9	157.6	203.6	248.6	293.5	348.5	4.63	0.364	<0.001	0.212
Daily milk yield (ECM kg)	28.1 ^A^	16.7 ^B^	28.6 ^A,b^	30.8 ^A,a^	28.7 ^A,b^	25.2 ^B^	22.0 ^B,C^	18.9 ^B^	15.8 ^B,C^	14.4 ^B,C^	0.43	<0.001	<0.001	0.031
Fat (%)	4.02 ^a^	3.86 ^b^	3.87	3.77	3.77	3.75	4.08	4.07	4.13	4.46	0.04	0.032	0.288	0.144
Protein (%)	3.43 ^a^	3.32 ^b^	2.99 ^b^	3.09 ^b^	3.17 ^b^	3.16 ^b^	3.30 ^a,b^	3.63 ^a^	3.57 ^a^	3.69 ^a^	0.07	0.044	0.029	0.581
Casein (%)	2.59	2.51	2.32	2.41	2.49	2.47	2.58	2.68	2.69	2.88	0.02	0.148	0.072	0.122
Lactose (%)	4.78 ^A^	4.61 ^B^	4.75	4.78	4.65	4.65	4.62	4.63	4.59	4.61	0.01	0.002	0.082	0.452
Dry matter (%)	12.76 ^A^	12.37 ^B^	12.28 ^B^	12.24 ^B^	12.24 ^B^	12.27 ^B^	12.69 ^A^	12.73 ^A^	12.74 ^A^	13.37 ^A^	0.05	<0.001	<0.001	0.144
Fat/protein ratio	1.23	1.21	1.3	1.22	1.22	1.19	1.22	1.21	1.21	1.22	0.15	0.503	0.643	0.611
SCC (10^3^ mL^−1^)	221.1	317.5	192.3	235.2	256.2	268.4	278.3	294.1	341.6	352.9	16.56			
Ln SCC	5.40 ^B^	5.76 ^A^	5.26	5.46	5.55	5.59	5.63	5.68	5.83	5.87	0.05	<0.001	0.879	0.777
Urea (mg L^−1^)	217.1 ^a^	196.2 ^b^	187.8	192.1	183.4	205.8	237	187.5	181.8	195.3	4.11	0.023	0.362	0.775
BCS	3.2 ^A^	3.0 ^B^	3.1 ^a,b^	2.7 ^b^	2.9 ^b^	3.0 ^a,b^	3.2 ^a,b^	3.3 ^a^	3.3 ^a^	3.4 ^a^	0.02	0.003	0.023	0.043

SCC—somatic cell count; BCS—body condition score; DIM—days in milk; CDH—conventional dairy herds; ODH—organic dairy herds; SE—standard error of the mean; means followed by different letters differ within rows (within the factor): A, B, C (*p* ≤ 0.01); a, b (*p* ≤ 0.05).

**Table 5 animals-14-03492-t005:** Effect of production system (PS) and lactation stage (LS) on the proportions and ratios of fatty acid groups in cow’s milk fat.

Item	Production System (PS)		Lactation Stage (LS) DIM								SE	*p*-Value		
	CDHs	ODHs	7–45	46–90	91–135	136–180	181–225	226–270	271–315	316–360		PS	LS	PSxLS
SCFAs	8.73 ^a^	8.34 ^b^	8.83 ^a^	8.83 ^a^	8.62 ^a^	8.54 ^a,b^	8.34 ^b^	8.28 ^b,c^	8.27 ^b,c^	8.05 ^c^	0.060	0.049	0.037	0.042
MCFAs	60.4	61.24	60.75	60.25	61.22	61.23	60.89	62.26	61.31	59.81	0.229	0.295	0.936	0.421
LCFAs	30.87	30.41	30.42	30.92	30.16	30.24	30.76	29.46	30.42	32.15	0.26	0.622	0.961	0.291
SFAs	65.96	66.17	66.05	65.75	66.53	65.94	66.06	67.2	66.57	64.48	0.261	0.852	0.919	0.214
UFAs	34.04	33.83	33.95	34.25	33.47	34.06	33.94	32.8	33.43	35.52	0.261	0.852	0.919	0.214
MUFAs	30.28	29.62	30.08	30.07	29.45	29.77	29.94	28.75	29.42	31.34	0.24	0.374	0.899	0.032
PUFAs	3.77 ^B^	4.21 ^A^	3.87	4.18	4.03	4.29	4.00	4.05	4.01	4.18	0.043	<0.001	0.406	0.876
n-3 PUFAs	0.58 ^B^	0.76 ^A^	0.64	0.68	0.7	0.8	0.65	0.74	0.68	0.74	0.015	<0.001	0.614	0.895
n-6 PUFAs	2.11	2.02	2.08	2.16	2.03	2.07	1.9	1.96	2.04	2.02	0.019	0.254	0.493	0.994
n-6/n-3 ratio	3.81 ^A^	3.02 ^B^	3.59	3.35	3.27	2.92	3.07	3.09	3.62	3.00	0.061	<0.001	0.389	0.833
AI	2.31	2.38	2.34	2.32	2.38	2.39	2.37	2.48	2.42	2.14	0.033	0.422	0.962	0.287
TI	2.89	2.89	2.9	2.83	2.93	2.86	2.85	3.02	3.00	2.69	0.035	0.986	0.882	0.338
HPI	0.45	0.43	0.45	0.44	0.42	0.43	0.41	0.41	0.42	0.48	0.007	0.507	0.634	0.046
Desaturase activity														
DI	0.092 ^B^	0.103 ^A^	0.097	0.095	0.094	0.108	0.112	0.101	0.094	0.108	0.002	0.007	0.486	0.324
Palmitoleic acid/palmitic acid	0.057	0.055	0.056	0.057	0.056	0.057	0.057	0.053	0.051	0.059	0.001	0.473	0.559	0.197
Oleic acid/stearic acid	2.05 ^b^	2.23 ^a^	2.08	2.24	2.16	2.4	2.21	2.17	2.04	2.13	0.032	0.045	0.525	0.278
CLA/vaccenic acid	0.47 ^B^	0.52 ^A^	0.49	0.5	0.49	0.57	0.5	0.53	0.52	0.47	0.007	0.001	0.597	0.364

DIM—days in milk; CDHs—conventional dairy herds; ODHs—organic dairy herds; SE—standard error of the mean; SCFAs—short-chain fatty acids; MCFAs—medium-chain fatty acids; LCFAs—long-chain fatty acids; SFAs—saturated fatty acids; UFAs—unsaturated fatty acids; MUFAs—monounsaturated fatty acids; PUFAs—polyunsaturated fatty acids; AI—atherogenicity index; TI—thrombogenic index; HPI—health-promoting index; DI—desaturase index. Means followed by different letters differ within rows (within the factor): A, B (*p* ≤ 0.01); a, b, c (*p* ≤ 0.05).

**Table 6 animals-14-03492-t006:** Effect of production system (PS) and lactation stage (LS) on the concentrations of functional fatty acids in cow’s milk fat.

Item	Production System (PS)		Lactation Stage (LS) DIM								SE	*p*-Value		
	CDHs	ODHs	7–45	46–90	91–135	136–180	181–225	226–270	271–315	316–360				
C4:0 (BA)	2.70 ^a^	2.55 ^b^	2.68 ^a^	2.68 ^a^	2.68 ^a^	2.58 ^ab^	2.51 ^b^	2.50 ^b^	2.48 ^b^	2.49 ^b^	0.02	0.047	0.054	0.918
C18:1 t10+11 (TVA)	1.36 ^B^	1.91 ^A^	1.52	1.75	1.67	1.73	1.97	1.83	1.6	2.07	0.046	<0.001	0.975	0.918
C18:1 c9 (OA)	23.32 ^a^	22.13 ^b^	22.99	22.57	22.15	22.34	22.26	21.49	22.53	23.67	0.216	0.045	0.916	0.053
C18:2 (LA)	1.96	1.88	1.93	2.01	1.89	1.92	1.78	1.82	1.89	1.90	0.018	0.282	0.568	0.978
C18:3 (LNA)	0.48 ^B^	0.65 ^A^	0.54	0.56	0.58	0.68	0.56	0.64	0.58	0.64	0.015	<0.001	0.768	0.901
C18:2 c9 t11 (CLA)	0.62 ^B^	0.95 ^A^	0.7	0.85	0.81	0.93	0.98	0.89	0.81	0.93	0.022	<0.001	0.921	0.94
C20:4 (AA)	0.15	0.14	0.15	0.15	0.14	0.15	0.12	0.14	0.15	0.12	0.002	0.297	0.371	0.233
C20:5 (EPA)	0.07 ^b^	0.08 ^a^	0.07	0.08	0.08	0.09	0.07	0.08	0.08	0.08	0.001	0.013	0.072	0.977
C22:5 (DPA)	0.12	0.13	0.12	0.12	0.12	0.14	0.11	0.12	0.13	0.11	0.002	0.069	0.381	0.175
C22:6 (DHA)	0.03	0.03	0.03	0.03	0.03	0.03	0.02	0.02	0.03	0.02	0.001	0.089	0.284	0.549

DIM—days in milk; CDHs—conventional dairy herds; ODHs—organic dairy herds; SE—standard error of the mean; BA—butyric acid; TVA—trans-vaccenic acid; OA—oleic acid; LA—linoleic acid; LNA—linolenic acid; CLA—conjugated linoleic acid; AA—arachidonic acid; EPA—eicosapentaenoic acid; DPA—docosapentaenoic acid; DHA—docosahexaenoic acid. Means followed by different letters differ within rows (within the factor): A, B (*p* ≤ 0.01); a, b (*p* ≤ 0.05).

**Table 7 animals-14-03492-t007:** Effect of production system (PS) and lactation stage (LS) on the concentrations of saturated fatty acids in cow’s milk fat.

Item	Production System (PS)		Lactation Stage (LS) DIM								SE	*p*-Value		
	CDHs	ODHs	7–45	46–90	91–135	136–180	181–225	226–270	271–315	316–360		PS	LS	PS × LS
C12:0 (LA)	3.35	3.24	3.3	3.42	3.23	3.2	3.58	3.26	3.33	3.07	0.043	0.399	0.841	0.298
C14:0 (MA)	11.03	11.18	11.07	11.18	11.12	11.11	11.66	11.32	11.28	10.57	0.087	0.601	0.998	0.384
C16:0 (PA)	28.67 ^a^	30.06 ^a^	29.06	29.02	29.98	30.69	29.00	30.57	29.57	28.79	0.219	0.047	0.717	0.331

DIM—days in milk; CDHs—conventional dairy herds; ODHs—organic dairy herds; SE—standard error of the mean; LA—lauric acid; MA—myristic acid; PA—palmitic acid; means followed by different letters differ within rows (within the factor): a (*p* ≤ 0.05).

## Data Availability

Datasets generated from the current experiment are available from the corresponding authors upon reasonable request.

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
