# Peer review of "The Health-Related Fatty Acid Profile of Milk from Holstein–Friesian Cows as Influenced by Production System and Lactation Stage"

_animals, 2024, doi:10.3390/ani14233492_

Round 1
Reviewer 1 Report
Comments and Suggestions for Authors
Dear authors,
your study is of interested for the highly discussed topic of differences between organic and conventional farming. The sample and laboratory design seem reasonable for me. But there is a need for major improvement of the rest of the manuscript. Please find my comments below:
Introduction:
L37 f. Reference Nr. 2 is rather vague here, Garbach talk about Probiotics like Lactobacilli, and not specifically about milk; bones and circulatory system are not mentioned in that paper. Please revise for a more accurate reference here.
L41: Reference Nr4 is not correct here, please find an appropriate reference for this sentence.
L44: I could not find Reference Nr. 6: but I am sure that there is many high impact papers about the environmental impact of dairy farms in general and organic vs. conventional farming, which could be cited here.
L47: I could not find Reference Nr. 9, please find another appropriate reference for this statement.
L48: Refr.Nr 10: Please give me the information where the review of Schwendel et al. talk about welfare and the connection to nutrition. I could not find this section in their manuscript.
Further: Schwendel et al. State in their review, that there is no clear difference in milk composition between conventional and organic farms. Please address this problem in your introduction. (whilst you stat in Line 45 ff that organic is regarded as healthier.)
L34 – 48: this section is rather weak, due to the bad referencing, and also lack of deepness. Please revise this section to make it more precise and scientific.
L60 f: RefrNr. 18: please cite the original article, that stated the origin of LCFAo or SMCFA; Nantapo is not he original author, and it is not a review.
L63 f: I would state this as “could potentially” decrease the risk for diabetes or cancer. There is no clinical trial that shows the clear anticancerogenic effect. If you know one, please cite this here.
L77: Please give me information where Refr.Nr. 27, Sun et al. state things like rumen capacity, passage rate, activity of SCD and so on, I could not find this information in their manuscript.
Materials and Methods:
L96: please state in what distance all the farms were, and if soil and grassland quality are similar for the ODH and CDH farms. So do they have similar geographic conditions? Are the ODH and CDH farms evenly distributed over this geographic region?
L96: Give information about the 305 days milk production of the farms used. Because just looking at the milk yield of 2 days per year does not tell anything about the milk yield in those farms. Also give the total number of cows on each farm, to show representativity of your samples.
L103: you repeat the information about the diet of the CDH here. Rearrange this paragraph.
Table 1 and 2: usually the contents of the diet are shown on a DM basis, in order to be able to compare two diets at first sight. Please revise.
L100 ff: Do you have more detailed information about the diets that were fed? How many kg of which diet component for CDH and ODH?
L115: please state how the 100mL of milk samples were taken (by hand or as separation milk during milking). How was the health status of the cows evaluated, was a CMT performed?
L151: Please state what the HPI means. Higher values mean a higher number of “healthy” FA? Where is the threshold where you speak of a health promotion effect?
L155 ff: were all other parameters normally distributed? What is your used alpha for significance? Also state that you analyzed the interaction between PS and LS;
Table 3: How can the mean lactation day in the group of 316-360 days be 368 days?
Fig 2: Was the BCS score performed by a trained person? And was there a second examiner who also scored the cows? If not, I would not put too much emphasis on BCS, and rather display other parameters in a graph, because BCS is a rather subjective measure, and should be validated by at least to examiners, to get a mean.
L192: this sentence seems a bit lost here.
Figure 3, and others: you did Tukeys Test as Posthoc, so pleas include superscripts for significant differences during the lactation days between ODH and CDH.
Discussion:
I did not check all your references in the discussion, but since there are serious flaws in the introduction, please re-read all your cited references of the discussion for correctness!
L 273f: please revise this sentence, it is wrong to state that organic farms cannot treat mastitis with antibiotics.
L385: what do you mean with “affected the course of lactation”? The conventional farms seem to have quite a normal lactation curve, where the organic farms show a rather fast decrease in milk yield, so low persistence. And I do not see a reflection of the fat mobilization in the milk curve. And following: L287: please show the statistics, that there is a significant difference for each the CDH and ODH between the different DIM; because the CDH were also increasing in BCS after the decline, so at that point there was no use of body fat reserves anymore.
L308: please give the measured concentrations of your diet here, to discuss the influence of the diet. You did the analysis, but you need to convert the numbers to g /DM ; otherwise you cannot compare the different feedstuff.
I would also recommend you, to link the dietary FA profile to the FA profile of the milk, in order to see correlations.
Table 4: Could you do the maths, if the ODH cows in total also produced more of the “favorable” FA per kg of milk? Because they had slightly higher proportions of for ex. CLA, but in total less Fat in the milk. So, as a consumer, maybe I still take up more favorable FA from milk from CDH, because they just have a higher Fat content. Please calculate this.
Minor comments:
Abstract
L20: Usually, the full breed name “Holstein Frisian” is used.
L29: Rephrase the last sentence, it is not elegant to read.
Table 3: 3,3 number of lactation (3.3); indicate the meaning of the superscripts; SCC: ths: does this mean thousand? Please rather use 103 ,
Is the daily milk yield (kg) the ECM? Or the actual yield? Where is the ECM?
Fig 2: use the same number of decimals for all numbers.
Table 5: headings are missing (P-value)
In general all figures and tables: check for complete headings, superscripts, footnotes etz.
Author Response
Responses to the Reviewers’
comments regarding Manuscript ID: animals-3285191 entitled “The health-related fatty acid profile of milk from Friesian cows as influenced by production system and lactation stage”
We are grateful to the Reviewers for thorough perusal of our manuscript and their valuable comments and suggestions, which helped us improve the overall quality of the paper.
The changes made are marked in red
Rev 1.
Introduction:
L37 f. Reference Nr. 2 is rather vague here, Garbach talk about Probiotics like Lactobacilli, and not specifically about milk; bones and circulatory system are not mentioned in that paper. Please revise for a more accurate reference here.
A: changed to Ratajczak et al., 2021
L41: Reference Nr4 is not correct here, please find an appropriate reference for this sentence.
A: changed to Thorning et al., 2016
L44: I could not find Reference Nr. 6: but I am sure that there is many high impact papers about the environmental impact of dairy farms in general and organic vs. conventional farming, which could be cited here.
A: [6] changed to Pirlo and Lolli, 2019
L47: I could not find Reference Nr. 9, please find another appropriate reference for this statement.
A: changed to Kasapidou et al., 2019
L48: Refr. Nr 10: Please give me the information where the review of Schwendel et al. talk about welfare and the connection to nutrition. I could not find this section in their manuscript.
A: changed to Spoolder, 2007
Further: Schwendel et al. State in their review, that there is no clear difference in milk composition between conventional and organic farms. Please address this problem in your introduction. (whilst you stat in Line 45 ff that organic is regarded as healthier.)
A: Schwendel et al. delated
L34 – 48: this section is rather weak, due to the bad referencing, and also lack of deepness. Please revise this section to make it more precise and scientific.
A: We changed some parts
L60 f: RefrNr. 18: please cite the original article, that stated the origin of LCFAo or SMCFA; Nantapo is not he original author, and it is not a review.
A: changed to Roca Fernandez and Gonzalez Rodriguez, 2012
L63 f: I would state this as “could potentially” decrease the risk for diabetes or cancer. There is no clinical trial that shows the clear anticancerogenic effect. If you know one, please cite this here.
A: The results of a cohort study suggest that high intakes of high-fat dairy foods and CLA may reduce the risk of colorectal cancer (Larsson, Bergkvist, & Wolk, 2005). Several studies in animals and humans have found an antidiabetic effect of CLA and suggested the trans-10,cis-12 isomer to be responsible for decreasing glucose levels and increased insulin sensitivity (Khanal, 2004)
[8] changed to Larsson et al., 2005; Khanal, 2004
L77: Please give me information where Refr. Nr. 27, Sun et al. state things like rumen capacity, passage rate, activity of SCD and so on, I could not find this information in their manuscript.
A: added Gross, 2022
Materials and Methods:
L96: please state in what distance all the farms were, and if soil and grassland quality are similar for the ODH and CDH farms. So do they have similar geographic conditions? Are the ODH and CDH farms evenly distributed over this geographic region?
A: added: The farms are located in the same Warmian-Masurian Voivodeship (north-eastern Poland). The quality of the soil was not tested. The quality of fodder (haylage and meadow hay) obtained from grasslands was examined.
L96: Give information about the 305 days milk production of the farms used. Because just looking at the milk yield of 2 days per year does not tell anything about the milk yield in those farms. Also give the total number of cows on each farm, to show representativity of your samples.
A: To characterize the herds in detail, two more tables should be built. The given daily yield in various phases of lactation (for several dozen animals in a herd) describes the level of milk productivity of herds.
L103: you repeat the information about the diet of the CDH here. Rearrange this paragraph.
A: Line 100 refers to all cows, and lines 103-105 refer to feeding systems. Table 1 and 2: usually the contents of the diet are shown on a DM basis, in order to be able to compare two diets at first sight. Please revise. A: converted into dry matter content
L100 ff: Do you have more detailed information about the diets that were fed? How many kg of which diet component for CDH and ODH?
A: We tested the composition of the feed. Added: Table 3. Ingredients (% dry matter, DM) and chemical composition of diets (these are averages for all herds)
L115: please state how the 100mL of milk samples were taken (by hand or as separation milk during milking). How was the health status of the cows evaluated, was a CMT performed?
A: added: Animals in all herds included in the study were under veterinary care.
changed to: Approximately 100 ml of milk was collected during milking using separators. Milk samples were collected in glass containers with 1 mL (20% wt/vol) of 2-bromo-2-nitropro-pane-1,3-diol (bronopol; VWR International AB, Stockholm, Sweden).
L151: Please state what the HPI means. Higher values mean a higher number of “healthy” FA? Where is the threshold where you speak of a health promotion effect?
A: added: HPI values range from 0.16 to 0.68. Dairy products with a high HPI value are assumed to be more beneficial to human health.
L155 ff: were all other parameters normally distributed? What is your used alpha for significance? Also state that you analyzed the interaction between PS and LS;
A: added: The remaining analyzed variables had a normal distribution.and: The significance level was set at p < 0.05 and at p < 0.01.
Information about taking into account the interaction between PS and LS is provided in the model formula: Yijk = μ + Ai + Bj + (AB)ij + eijk
Table 3: How can the mean lactation day in the group of 316-360 days be 368 days?
A: there was an error, it has been corrected
Fig 2: Was the BCS score performed by a trained person? And was there a second examiner who also scored the cows? If not, I would not put too much emphasis on BCS, and rather display other parameters in a graph, because BCS is a rather subjective measure, and should be validated by at least to examiners, to get a mean.
A: I assessed the BCS personally (Z. Nogalski). I assure you that I have experience (I teach others this method of assessing the level of energy reserves, I am the author of several works in the field of BCS).Added: were estimated by an experienced evaluator
L192: this sentence seems a bit lost here.
A: Yes, maybe a little lost. But the chart requires some explanation.
Figure 3, and others: you did Tukeys Test as Posthoc, so pleas include superscripts for significant differences during the lactation days between ODH and CDH.
A: corrected
I did not check all your references in the discussion, but since there are serious flaws in the introduction, please re-read all your cited references of the discussion for correctness!
A: corrected
L 273f: please revise this sentence, it is wrong to state that organic farms cannot treat mastitis with antibiotics.
A: changed to: SCC, which is strongly associated with udder health, tends to be higher in organic farms where the use of antibiotics to treat mastitis is highly limited.
L385: what do you mean with “affected the course of lactation”? The conventional farms seem to have quite a normal lactation curve, where the organic farms show a rather fast decrease in milk yield, so low persistence. And I do not see a reflection of the fat mobilization in the milk curve. And following: L287: please show the statistics, that there is a significant difference for each the CDH and ODH between the different DIM; because the CDH were also increasing in BCS after the decline, so at that point there was no use of body fat reserves anymore.
A: (it's about line 286) Mobilization of fat reserves (decrease in BCS) in CDH cows influenced the lactation curve. This did not occur in ODH cows (BCS unchanged, lactation without visible apex), as seen in Figures 1 and 2.
L308: please give the measured concentrations of your diet here, to discuss the influence of the diet. You did the analysis, but you need to convert the numbers to g /DM ; otherwise you cannot compare the different feedstuff.
I would also recommend you, to link the dietary FA profile to the FA profile of the milk, in order to see correlations.
A: added: The diets fed to cows in ODH were based on haylage, which is rich in linolenic acid (44.51%), and the diets fed to cows in CDH were based on ground grain and maize silage, which are rich in oleic acid (25.32% and 36.35%, respectively) and linoleic acid (46.38% and 27.63%, respectively).
Table 4: Could you do the maths, if the ODH cows in total also produced more of the “favorable” FA per kg of milk? Because they had slightly higher proportions of for ex. CLA, but in total less Fat in the milk. So, as a consumer, maybe I still take up more favorable FA from milk from CDH, because they just have a higher Fat content. Please calculate this.
A: added: These differences were due to differences in the diets, mainly in fiber content, which was higher in ODH (28.47%) than in CDH (19.17%). When comparing the FA profile of milk in both production systems and its impact on consumer health, milk fat content (4.025 in CDH and 3.86% in ODH) should be considered. For example, taking into account the differences in milk fat content, CLA concentration (0.62% in CDH and 0.95% in ODH) is 0.249 g and 0.366 g per L milk in CDH and ODH, respectively, which favors organic milk.
Minor comments:
Abstract
L20: Usually, the full breed name “Holstein Frisian” is used.
A: corrected
L29: Rephrase the last sentence, it is not elegant to read.
A: corrected to: Milk yield decreased (the decrease was more pronounced in ODH), the concentrations of milk constituents increased, and the proportion of short-chain fatty acids in milk fat decreased (p<0.05) with advancing lactation.
Table 3: 3,3 number of lactation (3.3); indicate the meaning of the superscripts; SCC: ths: does this mean thousand? Please rather use 103 ,
Is the daily milk yield (kg) the ECM? Or the actual yield? Where is the ECM?
A: corrected
Fig 2: use the same number of decimals for all numbers.
A: 0.25 was used because the BCS evaluation method uses an evaluation accuracy of 0.25
Table 5: headings are missing (P-value)
A: corrected
In general all figures and tables: check for complete headings, superscripts, footnotes etz.
A: corrected
Reviewer 2 Report
Comments and Suggestions for Authors
MANUSCRIPT TITLE: The health-related fatty acid profile of milk from Friesian cows as influenced by production system and lactation stage
MANUSCRIPT IDENTIFICATION: 3285191
SIMPLE SUMMARY
§ General considerations: Although a basis for the article's topic is necessary, I believe it can be done in a more objective way and not in 5 lines. This way, it will be possible to go into more depth and detail regarding the results obtained in this study that took into account non-specific data. only fatty acid profile, but also production and centesimal composition. Another important point is that in the abstract and in the material and methods of the article, 2 types of breeding systems were compared, organic and conventional. Therefore, I suggest that they be included the 2 types of abbreviations (ODH and CDH) in simple summary .
§ Lines 15 and 16: Which concentrations of milk constituents were increased in this study?
§ Line 17: What changes were observed in the milk fatty acid profile ? I suggest describing objectively and elaborating more fully the conclusion of the article, also taking into account the production and centesimal composition of the milk.
ABSTRACT
§ Line 19: I suggest replace “The aim of this study was to evaluate” with “This study aimed to evaluate”
§ Line 19: I suggest inserting “LS” after mentioning lactation stage .
§ Objective: I suggest reorganizing the order in which the variables evaluated in the article are presented according to the chronological order of the activities carried out in the field phase of the experiment. First, there is an evaluation of milk production, and then samples are collected for subsequent evaluations of the centesimal composition and fatty acid profile. Therefore, the information needs to be reorganized in the summary results section.
§ Lines 19 and 20: I suggest including in the objectives that the production, centesimal composition and fatty acid profile in milk were evaluated instead of mentioning “in particular the fatty acid profile of milk fat ”.
§ Line 21: I suggest including a more detailed description of the animals used in this study (weight and days in lactation with respective standard deviations, as well as the breed of cows).
§ Lines 24 and 25: I suggest inserting before proximate chemical composition, milk yield , since it was also evaluated in this study.
§ Line 24: I suggest replacing the expression “proximate chemical composition” by “proximate composition ”.
§ Lines 25 and 26: I suggest breaking the information into smaller sentences. First mention that average daily milk yield was higher in CHD than in ODH. And then mention that similarly the concentrations of fat, protein and lactose were higher in CHD than in ODH.
§ Line 27: must include the acronym PUFA after polyunsaturated fatty acids (PUFAs), so that then in line 28 the acronym for n-3 PUFAs can be used .
§ Line 28: I suggest removing the words “acid” after trans-vaccenic and linolenic and keeping only after linoleic mentioning it as “ acids ” instead of “ acid ”.
§ What is the conclusion of the work? Lines 29 and 30 described results that included P value for the reduction of milk fat content with advancing lactation. However, what are the conclusions regarding the effects of the systems evaluated on the variables analyzed?
KEYWORDS
§ I suggest organizing the keywords in alphabetical order;
§ bovine ” to the keywords , since the word “ cows ” has already been used in the title;
§ I suggest replacing the keyword “ desaturase index” with “ proximate composition ”.
INTRODUCTION
§ Lines 34 to 70: I suggest dividing the single paragraph into smaller paragraphs for better understanding of the information.
§ Line 40: I suggest mentioning the energy, mineral and vitamin needs of children with values, instead of stating that milk should contain an “abundant” quantity.
§ Line 43: What are the environmental impacts associated with milk production through conventional animal farming? I suggest describing this in more detail, as well as mentioning why conventional products are considered healthier compared to conventional animal farming (Lines 44 and 45).
§ Line 46: What would be considered “a more desirable fatty acid profile”? Which fatty acids are considered healthy in milk.
§ Line 49: What does EU mean ?
§ Lines 54 and 55: How does the milk produced become more nutritious and why does it have a more desirable fatty acid profile?
§ Line 59: I believe the acronyms for short and medium chain fatty acids need to be mentioned separately, SCFAs and MCFAs rather than as a single acronym.
§ Line 65: I suggest putting the acronym AI in parentheses.
§ Line 69: what would be “are more desirable ”?
§ Lines 89 and 90: I suggest standardizing the writing of the work objectives in the way they are mentioned in the summary.
§ I suggest that the authors add after or before the study objectives what the hypothesis of the work is. Do cows raised in an organic system have better or worse production, composition and fatty acid profile than animals raised in conventional systems? In addition, how is the lactation stage associated with changes in milk in cows raised in a conventional or traditional system?
MATERIAL AND METHODS
§ Line 93: I suggest adding information about the location where the experiment will be carried out to the animals topic. Besides, its necessary to include the experimental design used to distribute the animals in the treatments.
§ Line 94: I suggest including a more detailed description of the cows used in this study in relation to weight, days in lactation, with respective standard deviations, as well as the age of the animals, as previously suggested in the summary corrections.
§ Lines 96 and 97: I suggest adding more details about tie-stall and free-stall .
§ Table 1 caption: I suggest replacing the expression “ proximate chemical composition ” by “ proximate composition ”.
§ I suggest combining the information from tables 1 and 2 into the same table, since they refer to the same assessments. The chemical composition in both tables needs to be expressed in dry matter basis, not in on a fresh basis.
§ I suggest replacing the word “ specification ” with “item”.
§ I suggest correcting the spelling of “ silage ” to “ Maize silage ”.
§ The ground grain used refers to which feed ?
§ Why was crude fiber evaluated in this study and not mentioned neutral and acid detergent fiber, since cows, which are ruminant animals, were used? I suggest reviewing this information.
§ Why were only some fatty acids in foods evaluated? A complete evaluation of the fatty acid profile in foods would allow a better understanding of the effects of foods on the fatty acid profile in milk and on the production and centesimal composition of milk.
§ Line 100: What is the composition of the concentrates used in this study?
§ Line 101: I suggest replacing the expression “proximate chemical composition” by “proximate composition”.
§ Table 2: I suggest replacing the expression “proximate chemical composition” by “proximate composition”.
§ I suggest replacing the word “ specification ” with “item”.
§ I suggest correcting the spelling of “ silage ” to “ Maize silage ”.
§ Lines 108 to 123: I suggest rewriting the paragraph, breaking it down into smaller paragraphs to better understand the information. In addition, the font size used is different between lines.
§ Lines 110 to 112: how many days were used for data collection. I suggest including this information in addition to mentioning that it was done between the months of January and March.
§ Before mentioning the evaluations of milk composition, I suggest that it be mentioned that milk production was evaluated, with the respective methodology used.
§ Line 121: What does SCC mean?
§ Lines 131 to 152: I suggest dividing the paragraph into smaller paragraphs to improve understanding of the information.
§ Lines 131 and 132: Was the total dry extract content not evaluated in the milk samples? I suggest inserting this data, and also the information on defatted dry extract in the milk samples.
§ Lines 135 to 138: What types of gases are used in the chromatograph , how long was the device adjusted at each temperature?
§ Lines 139 to 141: What standards were used in this study? I suggest including this information in the article and brand used.
§ Lines 142 and 143: I suggest correcting the writing of the symbols before Celcius , only to “°” instead of “ º”.
§ Line 157: What is the meaning of Ln ?
§ Line 161: I suggest adding the acronym LS after lactation stage .
§ Line 164: I suggest putting the letters i, j, ij and eijk in italics.
§ Line 165: I suggest putting the letters ijk , i, j, ij and eijk in italics.
§ Line 168: What significance level was used for the variables in this study? Less than 5 %? I suggest including this information in the article.
RESULTS
§ Lines 171 and 172: I suggest removing this information from the article because it is unnecessary.
§ Line 172: replace “p” with “P” in (p<0.01).
§ Lines 174 and 175: What was the significant effect of lactation stage on milk production and composition?
§ Figure 1: I suggest removing the horizontal gray lines from the graph to make the figure less cluttered. In addition, it is necessary to insert black lines on the x and y axis in the figure to make it stand out more.
§ Line 180: I suggest inserting the acronym “LS” after lactation stage .
§ Lines 181 and 187: I suggest dividing the paragraphs according to the type of result and figure being evaluated. First mention all the results of figure 1 and in another paragraph, the result of figure 2. The way it was written made understanding the information confusing.
§ Figure 2: I suggest removing the horizontal gray lines from the graph to make the figure less cluttered. In addition, it is necessary to insert black lines on the x and y axis in the figure to make it stand out more.
§ Line 190: I suggest inserting the acronym “LS” after lactation stage .
§ Lines 192 and 193: Does the information mentioned refer to Figure 2? I suggest inserting the callout at the end of the sentence.
§ Table 3: I suggest inserting the acronym “LS” after lactation stage.
§ I suggest replacing all P-values described in Table 3 as “0.000” with “<0.001”.
§ fat / protein” was not mentioned in the material and methods. ratio ”.
§ What does “ ths ” mean in SCC ( ths mL-1)? Besides, it is necessary to include the P-values for this variable.
§ What is the meaning of Ln in “ Ln SCC”.
§ I suggest replacing “ scoring ” with “score” in the “BCS Condition Scoring ”.
§ I suggest inserting the assessed significance level in the table footer, as well as indicating that the averages were compared using the Tukey test , as was done in the footer of table 4.
§ Table 4: It is necessary to mention that the evaluation was carried out on cow's milk, as described in the statement in Table 3. In addition, I suggest replacing all P-values described in the table as “0.000” with “<0.001”.
§ Table 5: it is necessary to mention that the evaluation was carried out on cow's milk, as described in the statement in table 3.
§ Table 6: What are the meanings of the fatty acids evaluated? C12:0, C14:0 and C16:0. I suggest adding this information as was done with the other fatty acids in the previous table.
§ Table 6: in variable C16:0 in lactation stage 181-225, you need to standardize the decimal places as was done with the other averages.
§ Line 209: The authors mentioned that SCC increased with increasing lactation stage, however the differences were not significant. This information should not be mentioned in the results since it implies that there was no statistically significant effect.
§ Line 219: How was the proportion of SCFAs affected by lactation stage?
§ Line 233: Which fatty acids were influenced and how ?
§ Line 237: What does BA content mean ?
§ Line 238: What were the 3 fatty acids analyzed?
DISCUSSION
§ In general, I suggest that the authors make corrections to the writing of the discussion with the aim of making the paragraphs with fewer lines. This will make it possible to better understand the information and understand how both the lactation stage and the production systems can influence the quality of cows' milk.
I also suggest adding topics for milk production, composition and fatty acid profile to better understand the information.
§ Lines 244 to 261: As previously mentioned, I suggest that the paragraph be written in smaller paragraphs for better understanding of the information.
§ Lines 273 to 300: As previously mentioned, I suggest that the paragraph be written in smaller paragraphs for better understanding of the information.
§ Lines 301 to 342: As previously mentioned, I suggest that the paragraph be written in smaller paragraphs for better understanding of the information.
CONCLUSION
§ Although several significant results were verified, it is essential that the authors describe more objectively how the lactation stage and production systems can exert effects on the production, composition and profile of fatty acids in milk.
§ Therefore, the conclusion of the work needs to respond to the objective of the work and/or the hypothesis that is being tested.
REFERENCES
§ Standardize the font size of references according to the journal's standards.
§ 52 references were used in the article, however only 5 were mentioned from the last 5 years (between 2019 and 2024). Therefore, I suggest that an update be made to the introduction and discussion of the work in search of more recent works regarding the effect of the lactation stage and production systems on the quality of cow's milk.
FINAL DECISION
In view of the manuscript submitted by the authors, I suggest that it be revised again, with major corrections being essential, from the preparation of the abstract to the conclusions, as well as in the references. Therefore, although corrections have already been made by the authors, for the article to be suitable for acceptance, the information previously mentioned needs to be verified and corrected.
Author Response
Responses to the Reviewers’
comments regarding Manuscript ID: animals-3285191 entitled “The health-related fatty acid profile of milk from Friesian cows as influenced by production system and lactation stage”
We are grateful to the Reviewers for thorough perusal of our manuscript and their valuable comments and suggestions, which helped us improve the overall quality of the paper.
The changes made are marked in red
Rev 2
- Lines 15 and 16: Which concentrations of milk constituents were increased in this study?
A: protein and dry matter
- Line 17: What changes were observed in the milk fatty acid profile ? I suggest describing objectively and elaborating more fully the conclusion of the article, also taking into account the production and centesimal composition of the milk.
A: corrected
Line 19: I suggest replace “The aim of this study was to evaluate” with “This study aimed to evaluate”
A: it changed
- Line 19: I suggest inserting “LS” after mentioning lactation stage .
A: corrected
- Objective: I suggest reorganizing the order in which the variables evaluated in the article are presented according to the chronological order of the activities carried out in the field phase of the experiment. First, there is an evaluation of milk production, and then samples are collected for subsequent evaluations of the centesimal composition and fatty acid profile. Therefore, the information needs to be reorganized in the summary results section.
- Lines 19 and 20: I suggest including in the objectives that the production, centesimal composition and fatty acid profile in milk were evaluated instead of mentioning “in particular the fatty acid profile of milk fat ”.
A: changed
- Line 21: I suggest including a more detailed description of the animals used in this study (weight and days in lactation with respective standard deviations, as well as the breed of cows).
A: The breed is indicated in the title and purpose of work.
Information about the day of lactation is given in the next sentence. Cow weight was not assessed, only BCS.
- Lines 24 and 25: I suggest inserting before proximate chemical composition, milk yield , since it was also evaluated in this study.
A: added: (28.1 in CDH and 16.7 kg in ODH)
- Line 24: I suggest replacing the expression “proximate chemical composition” by “proximate composition ”.
A: changed
- Lines 25 and 26: I suggest breaking the information into smaller sentences. First mention that average daily milk yield was higher in CHD than in ODH. And then mention that similarly the concentrations of fat, protein and lactose were higher in CHD than in ODH.
A: changed
- Line 27: must include the acronym PUFA after polyunsaturated fatty acids (PUFAs), so that then in line 28 the acronym for n-3 PUFAs can be used .
A: corrected
- Line 28: I suggest removing the words “acid” after trans-vaccenic and linolenic and keeping only after linoleic mentioning it as “ acids ” instead of “ acid ”.
A: corrected
- What is the conclusion of the work? Lines 29 and 30 described results that included P value for the reduction of milk fat content with advancing lactation. However, what are the conclusions regarding the effects of the systems evaluated on the variables analyzed?
A: In the previous sentences, it was written about the influence of the lactation phase and the system on daily milk yield and fatty acid profile.
- I suggest organizing the keywords in alphabetical order;
A: corrected
- bovine ” to the keywords , since the word “ cows ” has already been used in the title;
A: added
- I suggest replacing the keyword “ desaturase index” with “ proximate composition ”.
A: corrected
Lines 34 to 70: I suggest dividing the single paragraph into smaller paragraphs for better understanding of the information.
A:corrected
- Line 40: I suggest mentioning the energy, mineral and vitamin needs of children with values, instead of stating that milk should contain an “abundant” quantity.
A: Milk plays a particularly important role in the growth and development of children whose diets should be abundant in nutrients, energy, minerals, and vitamins [3].
changed to: Milk consumption is associated with reduced stunting, and the World Health Organization (WHO) recommends that 25–33% of dietary protein should come from dairy products in malnourished children [3,4].
- Line 43: What are the environmental impacts associated with milk production through conventional animal farming? I suggest describing this in more detail, as well as mentioning why conventional products are considered healthier compared to conventional animal farming (Lines 44 and 45).
A: added: In organic farming synthetic fertilizers are banned, animal operations without land are not permitted and there are strict constrains to stocking rate.
- Line 46: What would be considered “a more desirable fatty acid profile”? Which fatty acids are considered healthy in milk.
A: added: organic milk contained more polyunsaturated fatty acids (PUFAs), including omega-6 and omega-3, than conventional milk.
- Line 49: What does EU mean ?
A: corrected
- Lines 54 and 55: How does the milk produced become more nutritious and why does it have a more desirable fatty acid profile?
A: Earlier it was written about feed, fertilizers, antibiotics...
- Line 59: I believe the acronyms for short and medium chain fatty acids need to be mentioned separately, SCFAs and MCFAs rather than as a single acronym.
A: corrected
- Line 65: I suggest putting the acronym AI in parentheses.
A: corrected
- Line 69: what would be “are more desirable ”?
A: Previous research has demon-68 strated that diets with a lower n–6/n–3 ratio are more desirable because they reduce the 69 risk of many chronic diseases [21].
Changed to: Previous research has shown that lowering the dietary n–6/n–3 ratio can reduce the risk of many chronic diseases [23].
- Lines 89 and 90: I suggest standardizing the writing of the work objectives in the way they are mentioned in the summary.
A: corrected
- I suggest that the authors add after or before the study objectives what the hypothesis of the work is. Do cows raised in an organic system have better or worse production, composition and fatty acid profile than animals raised in conventional systems? In addition, how is the lactation stage associated with changes in milk in cows raised in a conventional or traditional system?
A: corrected
Line 93: I suggest adding information about the location where the experiment will be carried out to the animals topic. Besides, its necessary to include the experimental design used to distribute the animals in the treatments.
A: added: The farms are located in the same Warmian-Masurian Voivodeship (north-eastern Poland).
It's written: The samples were collected randomly from up to 30% of cows at a given stage of lactation (7-45, 46-90, 91-135, 136-180, 181-225, 226-270, 271-315, and 316-360 days after calving).
- Line 94: I suggest including a more detailed description of the cows used in this study in relation to weight, days in lactation, with respective standard deviations, as well as the age of the animals, as previously suggested in the summary corrections.
A: In Table 4 it’s written: Number of lactations and Lactation day. The cows were not weighed.
- Lines 96 and 97: I suggest adding more details about tie-stall and free-stall .
A: added: In tie-stalls (with straw bedding) manure is removed mechanically twice a day before milking (chain scraper or manually). Pipeline milking machine, twice a day. In free-stall (with straw bedding) manure is removed mechanically once a day in the morning before feeding (tractor scraper). Milking twice a day (parlor auto tandem, fishbone milking parlor).
- Table 1 caption: I suggest replacing the expression “ proximate chemical composition ” by “ proximate composition ”.
A: changed
- I suggest combining the information from tables 1 and 2 into the same table, since they refer to the same assessments. The chemical composition in both tables needs to be expressed in dry matter basis, not in on a fresh basis.
A: I'm afraid it would be too large and less readable.
- I suggest replacing the word “ specification ” with “item”.
A: corrected
- I suggest correcting the spelling of “ silage ” to “ Maize silage ”.
A: It is written Maize silage
- The ground grain used refers to which feed ?
A: The ground grain used refers to the concentrate
- Why was crude fiber evaluated in this study and not mentioned neutral and acid detergent fiber, since cows, which are ruminant animals, were used? I suggest reviewing this information.
A: Thank You, we didn't analyze it.
- Why were only some fatty acids in foods evaluated? A complete evaluation of the fatty acid profile in foods would allow a better understanding of the effects of foods on the fatty acid profile in milk and on the production and centesimal composition of milk.
A: We agree with the Reviewer, we evaluated basic acids.
- Line 100: What is the composition of the concentrates used in this study?
A: added: The concentrate feed included cereal grains, corn, legume meal and rapeseed meal.
- Line 101: I suggest replacing the expression “proximate chemical composition” by “proximate composition”.
A: corrected
- Table 2: I suggest replacing the expression “proximate chemical composition” by “proximate composition”.
A: corrected
- I suggest replacing the word “ specification ” with “item”.
A: corrected
- I suggest correcting the spelling of “ silage ” to “ Maize silage ”.
A: It is written Maize silage
- Lines 108 to 123: I suggest rewriting the paragraph, breaking it down into smaller paragraphs to better understand the information. In addition, the font size used is different between lines.
A: corrected
- Lines 110 to 112: how many days were used for data collection. I suggest including this information in addition to mentioning that it was done between the months of January and March.
A: It is written: A total of 539 milk samples were collected twice over the winter season, in January (during morning milking) and in March (during evening milking).
14 days in January and 14 days in March (2 x one day in each herd)
- Before mentioning the evaluations of milk composition, I suggest that it be mentioned that milk production was evaluated, with the respective methodology used.
A: We did not use the results officially conducted in the herd. During the trial milking, we took milk samples ourselves and recorded the yield.
- Line 121: What does SCC mean?
A: added: somatic cell count
- Lines 131 to 152: I suggest dividing the paragraph into smaller paragraphs to improve understanding of the information.
A: corrected
- Lines 131 and 132: Was the total dry extract content not evaluated in the milk samples? I suggest inserting this data, and also the information on defatted dry extract in the milk samples.
A: We determined the dry matter of milk (Table 4), but we did not determine defatted dry extract
- Lines 135 to 138: What types of gases are used in the chromatograph , how long was the device adjusted at each temperature?
A: Helium was used as the carrier gas (line 142 ). The temperature program has been completed in line 143
Added: initial oven temperature – 110ºC, maintained for 5 min, then increased to 240ºC at a rate of 3ºC/min, maintained at 240ºC for 10 min, then increased to 249ºC at a rate of 1ºC/min. The total time of a single analysis was 68 min.
- Lines 139 to 141: What standards were used in this study? I suggest including this information in the article and brand used.
A: added: (Sigma Aldrich, Bellefonte, PA, USA).
Standards of all 41 tested fatty acids were used. C 4:0 butyric; C 6:0 caproic ; C 7:0 ; C 8:0 caprylic; C 10:0 capric; C 10:1 caprolei; C 11:0; C 12:0 lauric; C 12:1; C 13:0 iso; C 13:0 tridecanoic; C 14:0 iso; C 14:0 myristic; C 14:1 myristoleic; C 15:0 iso; C 15:0 anteiso; C 15:0 pentadecanoic; C 16:0 iso; C 16:0 palmitic; C 16:1 palmitoleic; C 17:0 margaric; C 17:1 margaricoleic; C 18:0 stearic; C 18:1 T6+9 elaidic; C 18:1 T10+11 vaccenic; C 18:1 C9 oleic; C 18:1 C11; C 18:1 C12; C 18:1 C13; C 18:1 T16; C 18:2 C9 T13; C 18:2 linoleic; C 18:3 α-linolenic; C 18:2 C9 T11 (CLA); C 20:1 gadoleic; C 20:2 eicosadienoic; C 20:4 arachidonic; C 22:0 behenic; C 20:5 eicosapentaenoic (EPA); C 22:5 docosapentaenoic (DPA); C 22:6 docosahexaenoic (DHA).
- Lines 142 and 143: I suggest correcting the writing of the symbols before Celcius , only to “°” instead of “ º”.
A: corrected
- Line 157: What is the meaning of Ln ?
A:it is Natural logarithm, ln is an abbreviation commonly used
- Line 161: I suggest adding the acronym LS after lactation stage .
A: corrected
- Line 164: I suggest putting the letters i, j, ij and eijk in italics.
A: corrected
- Line 165: I suggest putting the letters ijk , i, j, ij and eijk in italics.
A: corrected
- Line 168: What significance level was used for the variables in this study? Less than 5 %? I suggest including this information in the article.
A: added: The significance level was set at p < 0.05 and at p < 0.01.
Lines 171 and 172: I suggest removing this information from the article because it is unnecessary.
A: corrected
- Line 172: replace “p” with “P” in (p<0.01).
A: corrected
- Lines 174 and 175: What was the significant effect of lactation stage on milk production and composition?
A: added: In the later stages of lactation, daily milk yield decreased, and the concentrations of protein and dry matter in milk increased.
- Figure 1: I suggest removing the horizontal gray lines from the graph to make the figure less cluttered. In addition, it is necessary to insert black lines on the x and y axis in the figure to make it stand out more.
A: corrected
- Line 180: I suggest inserting the acronym “LS” after lactation stage .
A: corrected
- Lines 181 and 187: I suggest dividing the paragraphs according to the type of result and figure being evaluated. First mention all the results of figure 1 and in another paragraph, the result of figure 2. The way it was written made understanding the information confusing.
A: Further features are described in Table 4. The interaction is an extension of the description of daily milk yield and BCS.
- Figure 2: I suggest removing the horizontal gray lines from the graph to make the figure less cluttered. In addition, it is necessary to insert black lines on the x and y axis in the figure to make it stand out more.
A: corrected
- Line 190: I suggest inserting the acronym “LS” after lactation stage .
A: corrected
- Lines 192 and 193: Does the information mentioned refer to Figure 2? I suggest inserting the callout at the end of the sentence.
A: corrected
- Table 3: I suggest inserting the acronym “LS” after lactation stage.
A: corrected
- I suggest replacing all P-values described in Table 3 as “0.000” with “<0.001”.
A: corrected
- fat / protein” was not mentioned in the material and methods. ratio ”.
A: added: In addition, the fat/protein ratio was calculated
- What does “ ths ” mean in SCC ( ths mL-1)? Besides, it is necessary to include the P-values for this variable.
A:changed to 103 mL-1
In order to obtain the normal distribution of a variable, SCC was log-transformed
- What is the meaning of Ln in “ Ln SCC”.
A: Ln is the natural logarithm of SCC
- I suggest replacing “ scoring ” with “score” in the “BCS Condition Scoring ”.
A: corrected
- I suggest inserting the assessed significance level in the table footer, as well as indicating that the averages were compared using the Tukey test , as was done in the footer of table 4.
A: corrected
- Table 4: It is necessary to mention that the evaluation was carried out on cow's milk, as described in the statement in Table 3. In addition, I suggest replacing all P-values described in the table as “0.000” with “<0.001”.
A: corrected
- Table 5: it is necessary to mention that the evaluation was carried out on cow's milk, as described in the statement in table 3.
A: corrected
- Table 6: What are the meanings of the fatty acids evaluated? C12:0, C14:0 and C16:0. I suggest adding this information as was done with the other fatty acids in the previous table.
A added: LA – Lauric acid; MA – Myristic acid; PA – Palmitic acid
- Table 6: in variable C16:0 in lactation stage 181-225, you need to standardize the decimal places as was done with the other averages.
A: corrected
- Line 209: The authors mentioned that SCC increased with increasing lactation stage, however the differences were not significant. This information should not be mentioned in the results since it implies that there was no statistically significant effect.
A: It was written: SCC increased with advancing lactation, but the differences were not significant.
No one is saying they were important. There is no mention of this in the abstract, conclusions
- Line 219: How was the proportion of SCFAs affected by lactation stage?
A: The proportion of SCFAs was affected by lactation stage, and it decreased with advancing lactation.
- Line 233: Which fatty acids were influenced and how ?
A: This is explained in the next two sentences.
- Line 237: What does BA content mean ?
A: added: Butyric acid
- Line 238: What were the 3 fatty acids analyzed?
A: added: (lauric, myristic and palmitic)
- Lines 244 to 261: As previously mentioned, I suggest that the paragraph be written in smaller paragraphs for better understanding of the information.
A: corrected
- Lines 273 to 300: As previously mentioned, I suggest that the paragraph be written in smaller paragraphs for better understanding of the information.
A: corrected
- Lines 301 to 342: As previously mentioned, I suggest that the paragraph be written in smaller paragraphs for better understanding of the information.
A: corrected
- Although several significant results were verified, it is essential that the authors describe more objectively how the lactation stage and production systems can exert effects on the production, composition and profile of fatty acids in milk.
- Therefore, the conclusion of the work needs to respond to the objective of the work and/or the hypothesis that is being tested.
A: corrected
- Standardize the font size of references according to the journal's standards.
A: corrected
- 52 references were used in the article, however only 5 were mentioned from the last 5 years (between 2019 and 2024). Therefore, I suggest that an update be made to the introduction and discussion of the work in search of more recent works regarding the effect of the lactation stage and production systems on the quality of cow's milk.
A: 11 new articles were introduced, including 5 from 2019-2024.
We would like to thank the reviewers once again for their effort in assessing our work.

Round 2
Reviewer 1 Report
Comments and Suggestions for Authors
Dear Authors,
thank you for considering my comments.
There is a few minor comments left:
L43 Ratajczak does not mention the circulatory system, I think you meant reference 4 to be also here at the end of this sentence? (Thorning?)
and then L45: remove Refr. 4 after this sentence, only 3 is correct here .
L44: I could not find Reference Nr. 6: but I am sure that there is many high impact papers about the environmental impact of dairy farms in general and organic vs. conventional farming, which could be cited here.
A: [6] changed to Pirlo and Lolli, 2019
R: Please cite the original reference here, which is the EU legislation (as Pirlo and Lolli did).
àL 72: it should be reference no. 20 and 21 or? (not 22)
L66; NEB in parentheses (NEB)
L96: Give information about the 305 days milk production of the farms used. Because just looking at the milk yield of 2 days per year does not tell anything about the milk yield in those farms. Also give the total number of cows on each farm, to show representativity of your samples.
A: To characterize the herds in detail, two more tables should be built. The given daily yield in various phases of lactation (for several dozen animals in a herd) describes the level of milk productivity of herds.
R: I meant as an indication when you describe the herds, what was their average herd milk yeald per year, how many cows were on those farms? Not for statistics. But to give the reader an idea, what size and production level your herds of the organic and conventional farms were.
L192: this sentence seems a bit lost here.
A: Yes, maybe a little lost. But the chart requires some explanation.
L 222 Yes, but make sure the sentence is connected to the paragraph, and not a stand-alone sentence. Just move it up to L 217.
Since there is still a bit of a mess with the Refrences, please revise all refrences carefully again. It is part of good scientific practice to do correct citing.
Author Response
Responses to the Reviewers’
comments regarding Manuscript ID: animals-3285191 entitled “The health-related fatty acid profile of milk from Holstein-Friesian cows as influenced by production system and lactation stage”
We are grateful to the Reviewers for re-reading our manuscript and their valuable comments and suggestions, which helped us improve the overall quality of the paper.
The changes made are marked in red
Rev. 1
Dear Authors,
thank you for considering my comments.
There is a few minor comments left:
R: L43 Ratajczak does not mention the circulatory system, I think you meant reference 4 to be also here at the end of this sentence? (Thorning?)
and then L45: remove Refr. 4 after this sentence, only 3 is correct here .
A: Written: “The consumption of milk and dairy products has a beneficial influence on bones, … „ ([2] reports about it;
[3] added next to [2], in [3] written: Consumption of dairy products has been shown to have a positive impact on bone mass, cardiovascular health ..
[4] corrected
R: L44: I could not find Reference Nr. 6: but I am sure that there is many high impact papers about the environmental impact of dairy farms in general and organic vs. conventional farming, which could be cited here.
A: [6] changed to Pirlo and Lolli, 2019
R: Please cite the original reference here, which is the EU legislation (as Pirlo and Lolli did).
A: added: the EU legislation [7]
R: L 72: it should be reference no. 20 and 21 or? (not 22)
A: corrected
R: L66; NEB in parentheses (NEB)
A: corrected
R: L96: Give information about the 305 days milk production of the farms used. Because just looking at the milk yield of 2 days per year does not tell anything about the milk yield in those farms. Also give the total number of cows on each farm, to show representativity of your samples.
A: added: The average number of cows in CDH was 52 and in ODH 44. The average annual milk yield in CDH was 9600 kg and in ODH 6200 kg.
R: L192: this sentence seems a bit lost here.
A: Yes, maybe a little lost. But the chart requires some explanation.
R: L 222 Yes, but make sure the sentence is connected to the paragraph, and not a stand-alone sentence. Just move it up to L 217.
A: corrected
R: Since there is still a bit of a mess with the Refrences, please revise all refrences carefully again. It is part of good scientific practice to do correct citing.
A: corrected

Reviewer 2 Report
Comments and Suggestions for Authors
Dear authors, respectfully, it seems to me that this is a good manuscript. However, many points affect the quality of the manuscript. The main factor that affects the manuscript is the generic writing style that decreases the relevance of the text. The novelty of the manuscript is not clear. Some suggestions are below.
Simple summary: The new information added is confusing to read. Write in a simple way, trying to describe the text to promote a fluid reading.
Abstract: similar to the simple summary, the description of the abstract is confusing to read. Try to describe the text to promote a fluid reading. Add p-values and write the results in a different way that a simple translation of the tables.
Line 24: Were the animals chosen at random? Add the statistical design followed.
Line 27: Add how the data was statistically analyzed.
Line 33: Add the conclusion of the study.
Introduction:
Reorganize the data in the introduction. Some paragraphs describe one topic, the next describes another topic, and the next describes the first topic, avoiding a logical sequence.
Avoid generic descriptions such as: animals have lower production. Describe in detail: Dairy cows on organic farms produced approximately 15 l/d; less than the 30 l/d produced by cows on non-organic farms.
I read the entire introduction and did not find the novelty of the manuscript, even though I read the hypothesis and the objective. Authors should highlight the novelty of the study in the introduction.
Material and methods
Place Table 1 after the in-text citation.
Results. The results description is generic. Improve your writing style showing your data in other forms more than a simple description of tables. Was higher? How much (%, g, l, etc)?
Discussion: The topic of discussion is very speculative. The discussion should focus on explaining how the results were obtained. In the current situation, the discussion is a comparison of data with other authors; however, you need to make a specific description of how the results were obtained.
Example of generic description: In the present study, milk fat content was lower in ODH, which could be due to inadequate feeding. What does “inadequate nutrition” mean? Decreased dietary intake, decreased nutrient intake? Decreased intake of a specific nutrient? Unbalanced diet? Etc.
Lines 275-276: CDH against CDH? This result is expected, there is no need to corroborate it. Instead of comparing your data with other authors, explain more specifically how you obtained your results. This suggestion applies not only to this statement, but to every discussion of this topic. Comparing your data to other authors' data is not an interesting way to discuss your data.
Conclusion: In the conclusion, add a paragraph indicating the positive and negative points of each production system considering the results of the study.
Author Response
Rev. 2
R: Simple summary: The new information added is confusing to read. Write in a simple way, trying to describe the text to promote a fluid reading.
A: added (in accordance with the Reviewers' suggestions) In the present study, milk yield decreased (the decrease was more pronounced in organic herds-ODH) and the concentrations of protein and dry matter increased with advancing lactation. The fatty acid profile of milk fat was more desirable in ODH than in CDH (higher concentrations of polyunsaturated fatty acids (PUFAs), including n-3 PUFAs, trans-vaccenic acid, linolenic acid, and conjugated linoleic acid, and a higher desaturase index).
- We don't know what's confusing to read here?
R: Abstract: similar to the simple summary, the description of the abstract is confusing to read. Try to describe the text to promote a fluid reading. Add p-values and write the results in a different way that a simple translation of the tables.
A: After corrections the text has been re-translated by a licensed translator. Your comments aim to return to the original version of the text. Otherwise, the p-value was reported twice. R: Line 24: Were the animals chosen at random? Add the statistical design followed.
A: added in: 2.1. Animals: Cows were selected in particular stages of lactation according to the order of calving.
R: Line 27: Add how the data was statistically analyzed.
A: The abstract is limited in size; statistics are given in section 2.3
R: Line 33: Add the conclusion of the study.
A: The research summary is contained in lines 27-33
R: Introduction:
R: Reorganize the data in the introduction. Some paragraphs describe one topic, the next describes another topic, and the next describes the first topic, avoiding a logical sequence.
A: The first paragraph talks about the importance of milk, and the next about its production in various systems.
R: Avoid generic descriptions such as: animals have lower production. Describe in detail: Dairy cows on organic farms produced approximately 15 l/d; less than the 30 l/d produced by cows on non-organic farms.
A: It's written: “Average daily milk yield was 11.4 kg higher (P<0.01) in CDH than in ODH (Table 4).”
R: I read the entire introduction and did not find the novelty of the manuscript, even though I read the hypothesis and the objective. Authors should highlight the novelty of the study in the introduction.
A: In the introduction it was written:The justification of the research and its originality are included in this part of the introduction: Most researchers investigated the effect of nutritional factors and focused mainly on the first few weeks after calving. However, the influence of lactation stage (LS) on the composition of milk fat has been rarely studied, and the available reports are often inconclusive [20,28,29]. Feed intake changes in different stages of lactation, which affects the content of enzymatic and microbial flora, rumen capacity, rumen passage rates, rumen isomerization, biohydrogenation, the activity of stearoyl-CoA desaturase (SCD) in the mammary gland, and energy balance in dairy cows [30,31]. At the beginning of lactation, body fat reserves in high-yielding cows are mobilized due to NEB, which leads to the release of LCFAs (mainly stearic and oleic acids) from adipocytes [32]. According to Stoop et al. [18], LS and the corresponding energy balance contribute significantly to the variation in milk fat composition by modifying the activity of FA pathways. The research hypothesis postulates that cows kept in organic herds and fed extensively with organic feed will produce less milk while remaining more stable during lactation than intensively fed cows that are at higher risk of NEB in the first stage of lactation. Therefore, this study aimed to evaluate the effect of production system and LS on the yield, centesimal composition and fatty acid profile of milk from Holstein-Friesian cows.
R: Material and methods
Place Table 1 after the in-text citation.
A: corrected
R: Results. The results description is generic. Improve your writing style showing your data in other forms more than a simple description of tables. Was higher? How much (%, g, l, etc)?
A: corrected
R: Discussion: The topic of discussion is very speculative. The discussion should focus on explaining how the results were obtained. In the current situation, the discussion is a comparison of data with other authors; however, you need to make a specific description of how the results were obtained.
A: .. speculative?” What does mean?
How we obtained the results is described in section 2. Materials and Methods. In discussion we focused on explaining the differences obtained, among others, by comparison with other studies.
R: Example of generic description: In the present study, milk fat content was lower in ODH, which could be due to inadequate feeding. What does “inadequate nutrition” mean? Decreased dietary intake, decreased nutrient intake? Decreased intake of a specific nutrient? Unbalanced diet? Etc
A: changed: In the present study, milk fat content was lower in ODH, which could be due to less abundant feeding and the absence of dietary supplements.
added: In conventional dairy farming, high-yielding cows are commonly fed in excess of requirements using diets supplemented with high amounts of energy and protein concentrates. However, in organic dairy farming the aim is to optimize available resources rather than maximize production, so that in most cases systems are based on the maximum use of forage [46].
R: Lines 275-276: CDH against CDH? This result is expected, there is no need to corroborate it. Instead of comparing your data with other authors, explain more specifically how you obtained your results. This suggestion applies not only to this statement, but to every discussion of this topic. Comparing your data to other authors' data is not an interesting way to discuss your data.
A: Comparing the results obtained with those of other authors is common in discussions. There are not many comparisons in our discussion. If there are any, it is in order to better explain the obtained differences.For example: In the present study, the SCC increased with advancing lactation, which could contribute to the decrease in milk production. According to Gonçalves et al. [49], an increase in the SCC is accompanied by a decrease in milk yield, especially when the SCC exceeds the threshold of 200 000 cells per mL of milk.orIn the current study, the difference in daily milk yield between the analyzed systems reached 11.4 kg, and BarÅ‚owska et al. [40] reported that Holstein-Friesian cows in the conventional system produced on average 6.1 kg more milk than their counterparts in the organic system. The higher milk yield in conventional farms is the result of both more intensive feeding and optimally balanced diets, compared with organic farms. Król et al. [41] demonstrated that the nutrient requirements of dairy cows were not fully met in the organic production system. Inadequate coverage of the nutritional needs of cows in organic herds is also the main reason for lower concentrations of milk constituents, relative to conventional herds [42].
R: Conclusion: In the conclusion, add a paragraph indicating the positive and negative points of each production system considering the results of the study.
A: This would be a repetition of the content contained in the first two sentences of the conclusions.
